# Defending against Model Stealing via Verifying Embedded External Features

**Linghui Zhu** [* 1]  **Yiming Li** [* 1]  **Xiaojun Jia** [2]  **Yong Jiang** [1]  **Shu-Tao Xia** [1]  **Xiaochun Cao** [2]

## Abstract

Well-trained models are valuable intellectual properties for their owners. Recent studies revealed that the adversaries can 'steal' deployed models even when they have no training sample and can only query the model. Currently, there were some defense methods to alleviate this threat, mostly by increasing the cost of model stealing. In this paper, we explore the defense problem from another angle by *verifying whether a suspicious model contains the knowledge of defender-specified external features*. We embed the *external features* by *poisoning* a few training samples via style transfer. After that, we train a meta-classifier, based on the gradient of predictions, to determine whether a suspicious model is stolen from the victim. Our method is inspired by the understanding that the stolen models should contain the knowledge of (external) features learned by the victim model. Experimental results demonstrate that our approach is effective in defending against different model stealing attacks simultaneously.

## 1. Introduction

Deep neural networks (DNNs) have been widely and successfully adopted in many areas. In general, training a well-performed model requires a large number of valuable resources therefore the trained model is a valuable intellectual property. Recently, researchers found that adversaries can 'steal' deployed (victim) models even when they have no training sample and can only query the model (Tramèr et al., 2016; Orekondy et al., 2019; Chandrasekaran et al., 2020). This threat is called *model stealing*. Since the model stealing can obtain a function-similar copy of the victim model stealthily, it poses a huge threat to model owners.

[*]Equal contribution  [1]Tsinghua Shenzhen International Graduate School, Tsinghua University [2]Institute of Information Engineering, Chinese Academy of Sciences. Correspondence to: Yiming Li <li-ym18@mails.tsinghua.edu.cn>, Shu-Tao Xia <xiast@sz.tsinghua.edu.cn>.

*Accepted by the ICML 2021 workshop on A Blessing in Disguise: The Prospects and Perils of Adversarial Machine Learning.*

To alleviate the threat of model stealing, there were also some defense methods, mostly by introducing randomness or perturbation in the victim models to increase the costs of model stealing (Tramèr et al., 2016; Lee et al., 2019; Kariyappa & Qureshi, 2020). For instance, defenders may perturb the prediction by rounding or adding noise to the posterior probabilities. However, these defenses may significantly reduce the performance of legitimate users and could even be bypassed by adaptive attacks (Maini et al., 2021).

In this paper, we explore the defense of model stealing from another perspective by *verifying whether a suspicious model has defender-specified behaviors*. If the model has such behaviors, we treat it as stolen from the victim. It is inspired by the understanding that the stolen models should contain the knowledge of features learned by the victim model therefore they have similar behaviors. To the best of our knowledge, there is only one work, the *dataset inference* (Maini et al., 2021), focusing on this perspective, where they adopted *inherent features* of the training set to verify model ownership. However, we reveal that this method is easy to make misjudgments, especially when the training set of suspicious models have a similar distribution to that of the victim model. Based on this observation, we propose to embed defender-specified *external features* into victim models for ownership verification. Specifically, we embed external features by *poisoning* some training samples via style transfer (Johnson et al., 2016). Since we only poison a few samples and do not change their labels, the embedded features will not hinder the functionality of the victim model. In the meanwhile, we also train a benign model based on the original training set. This model is used only for training a *meta classifier* to determine whether a suspicious model is stolen from the victim model. Only the model containing the knowledge of external features will be deployed.

In conclusion, the main contribution of this work is fourfold: **(1)** We revisit the defense of model stealing from the aspect of ownership verification. **(2)** We reveal the limitations of existing verification-based methods and propose a simple yet effective defense approach. **(3)** We verify the effectiveness of our method on benchmark datasets under various types of model stealing attacks simultaneously. **(4)** Our work could provide a new angle about how to adopt 'data poisoning' for positive purposes.

## 2. Related Work

### 2.1. Model Stealing

Model stealing aims to steal the intellectual property from a victim by obtaining a function-similar copy of the deployed model. In general, existing model stealing methods can be divided into three main categories based on the adversary's permission level, as follows:

**Dataset-Accessible Attacks ($\mathcal{A}_D$):** In this setting, the adversary can get access to the training dataset whereas can only query the model. The adversary can train a substitute model by the knowledge distillation (Hinton et al., 2014) or simply training their own model from the scratch.

**Model-Accessible Attacks ($\mathcal{A}_M$):** In this setting, the adversary has complete access to the victim model. The adversary can obtain a substitute model by using data-free distillation (Fang et al., 2019; Truong et al., 2021) or simply fine-tuning the victim model with local training samples.

**Query-Only Attacks ($\mathcal{A}_Q$):** In this setting, the adversary can only query the model. Specifically, the query-only attacks can also be divided into two sub-classes, including the *label-query attacks* (Papernot et al., 2017; Jagielski et al., 2020; Chandrasekaran et al., 2020) and *logit-query attacks* (Tramèr et al., 2016; Orekondy et al., 2019), based on the type of victim model's feedback (*e.g.*, label or probabilities).

### 2.2. Verification-based Defenses against Model Stealing

The verification-based methods against model stealing by conducting the ownership verification towards the suspicious model. To the best of our knowledge, there is currently only one research (*i.e.* dataset inference (Maini et al., 2021)) completely working on this area. Its key idea is to identify whether the suspicious model contains the knowledge of *inherent features* that the victim model $V$ learned from a private training set. Specifically, for each sample $(\boldsymbol{x}, y)$ in the $K$-classification problem, dataset inference first generated its minimum distance $\boldsymbol{\delta}_t$ to each class $t$ by

$$\min_{\boldsymbol{\delta}_t} d(\boldsymbol{x}, \boldsymbol{x} + \boldsymbol{\delta}_t), s.t., V(\boldsymbol{x} + \boldsymbol{\delta}_t) = t, \qquad (1)$$

where $d(\cdot)$ is a distance metric, such as the $\ell^\infty$ norm. The distance to each class $\boldsymbol{\delta} = (\boldsymbol{\delta}_1, \cdots, \boldsymbol{\delta}_K)$ is the feature embedding of sample $(\boldsymbol{x}, y)$ *w.r.t.* the victim model $V$. After that, the defender will randomly select some samples inside (labeled as '+1') or out-side (labeled as '-1') their private dataset and use their feature embedding $\boldsymbol{\delta}$ to train a binary meta-classifier. To determine whether a suspicious model $S$ is stolen from the victim, the defender creates equal-sized sample vectors from private and public samples and conduct the hypothesis test. If the confidence scores of private samples are significantly greater than those of public samples, the suspicious model $S$ is treated as stolen from the victim.

*Table 1.* p-value of verifications where dataset inference misjudges.

|  | ResNet-$\mathcal{D}_r$ | VGG-$\mathcal{D}_l$ | VGG-$\mathcal{D}_l'$ |
| --- | --- | --- | --- |
| p-value | $10^{-7}$ | $10^{-5}$ | $10^{-4}$ |

We notice that the dataset inference enjoys certain similarities to the *backdoor-based model watermarking* (Adi et al., 2018; Zhang et al., 2018; Li et al., 2020b), which makes these approaches to be potential defenses against model stealing. Specifically, these methods first adopted *backdoor attacks* (Li et al., 2020a) to watermark the model and then conducted the ownership verification based on the predictions of benign and poisoned samples. If the confidence scores in predicting the target label of poisoned samples are significantly greater than those of their benign version, the suspicious model is treated as watermarked and therefore it is stolen from the owner. Note that the original purpose of model watermarking is detecting theft (*i.e.*, directly copy the model) rather than preventing model stealing.

## 3. The Limitation of Existing Defenses

### 3.1. The Limitation of Dataset Inference

The dataset inference relied on a latent assumption that an independent model will not learn the features contained in the private dataset. However, this assumption may not hold and therefore the method may misjudge. Here we verify it.

**Settings.** We conduct experiments with VGG (Simonyan & Zisserman, 2015) and ResNet (He et al., 2016) on CIFAR-10 (Krizhevsky et al., 2009) dataset. Specifically, we split the original training set into two disjoint subsets $\mathcal{D}_l$ and $\mathcal{D}_r$. We train the VGG on $\mathcal{D}_l$ (dubbed VGG-$\mathcal{D}_l$) and the ResNet on $\mathcal{D}_r$ (dubbed ResNet-$\mathcal{D}_r$). We also train the VGG on a noisy dataset $\mathcal{D}_l' \triangleq \{(\boldsymbol{x}', y)|\boldsymbol{x}' = \boldsymbol{x} + \mathcal{N}(0, 16), (\boldsymbol{x}, y) \in \mathcal{D}_l\}$ (dubbed VGG-$\mathcal{D}_l'$) for reference. In the verification stage, we verify whether the VGG-$\mathcal{D}_l$ and VGG-$\mathcal{D}_l'$ is stolen from ResNet-$\mathcal{D}_r$ and whether the ResNet-$\mathcal{D}_r$ is stolen from VGG-$\mathcal{D}_l$ based on dataset inference (Maini et al., 2021). Besides, we also adopt p-value as the evaluation metric. *The smaller the p-value, the more confident that dataset inference believes the model stealing happened.*

**Results.** As shown in Table 1, all p-values are significantly smaller than 0.01, *i.e.*, dataset inference believes that these models are stolen from the victim. However, in each case, since the suspicious and victim model was trained on completely different training samples and with different structures, the suspicious model should not be considered as stolen from the victim. These results reveal that *dataset inference could make misjudgments and therefore its results are questionable*. In particular, the p-value of VGG-$\mathcal{D}_l$ is smaller than that of the VGG-$\mathcal{D}_l'$. This is probably because the latent distribution of $\mathcal{D}_l'$ is more different from that of $\mathcal{D}_r$ and therefore models learn more different features.

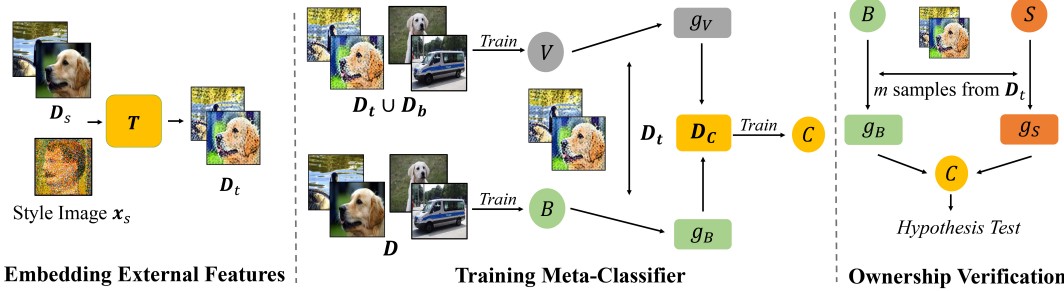

*Figure 1.* The main pipeline of our method. In the first stage, defenders will modify some images via the style transfer for embedding external features. In the second stage, defenders will train a meta-classifier to determine whether a suspicious model is stolen from the victim based on the gradients. In the last stage, defenders will conduct ownership verification with the hypothesis test.

*Table 2.* The performance (%) of different models.

| Model Type → | Benign | Watermarked | Stolen |
|---|---|---|---|
| BA | 91.99 | 85.49 | 70.17 |
| ASR | 0.01 | 100.00 | **3.84** |

### 3.2. The Limitation of Model Watermarking

Backdoor-based model watermarking relied on an assumption that the trigger matches hidden backdoors contained in the suspicious model. However, the assumption may not hold. In this section, we verify this limitation.

**Settings.** We adopt the most mainstream and effective backdoor attack, the BadNets (Gu et al., 2019), for the model watermarking. The watermarked model will then be stolen by the data-free distillation-based stealing attack (Fang et al., 2019). We adopt the *benign accuracy (BA)* and *attack success rate (ASR)* (Li et al., 2020a) for the evaluation. The larger the ASR, the better the trigger-backdoor match.

**Results.** As shown in Table 2, the ASR of the stolen model is only 3.84%, which is significantly lower than that of the watermarked model, *i.e.*, *the defender-specified trigger is no longer matches the hidden backdoors contained in the stolen model*. As such, backdoor-based model watermarking will has minor effects in defending against model stealing.

## 4. The Proposed Method

Based on the understandings in Section 3, in this paper, we propose to embed *external features* instead of the inherent features or a specific (trigger) pattern for ownership verification. Specifically, as shown in Figure 1, our method consists of three main stages, including **(1)** embedding external features, **(2)** training an ownership meta-classifier, and **(3)** conducting ownership verification. Their technical details are in the following subsections.

### 4.1. Threat Model

We consider the defense in a *white-box* setting, where the defender has complete access to the suspicious model. How-ever, the defender has no information about the stealing process. The goal of defenders is to accurately identify whether a suspicious model is stolen from a victim model.

### 4.2. Embedding External Features

Before we reach the technical details, we first present the definition of the inherent and external features.

**Definition 1.** *A feature $f$ is called the inherent feature of the dataset $\mathcal{D}$ if and only if $\forall (\boldsymbol{x}, y) \in \mathcal{X} \times \mathcal{Y}, (\boldsymbol{x}, y) \in \mathcal{D} \Rightarrow (\boldsymbol{x}, y)$ contains feature $f$. Similarly, $f$ is called the external feature (of the dataset $\mathcal{D}$) if and only if $\forall (\boldsymbol{x}, y) \in \mathcal{X} \times \mathcal{Y}, (\boldsymbol{x}, y)$ contains feature $f \Rightarrow (\boldsymbol{x}, y) \notin \mathcal{D}$.*

In this paper, we consider the classification of natural images. As such, defenders can easily embed external features via style transfer (Johnson et al., 2016) with a particular *style image* (*e.g.*, the painting). Specifically, let $\mathcal{D} = \{(\boldsymbol{x}_i, y_i)\}_{i=1}^{N}$ denotes the benign training set, $\boldsymbol{x}_s$ is a defender-specified style image, and $T : \mathcal{X} \times \mathcal{X} \to \mathcal{X}$ is a (trained) style transformer. In this stage, the defender first randomly selects $\gamma\%$ (dubbed *transformation rate*) samples (*i.e.*, $\mathcal{D}_s$) from $\mathcal{D}$ to generate their transformed version $\mathcal{D}_t = \{(\boldsymbol{x}', y) | \boldsymbol{x}' = T(\boldsymbol{x}, \boldsymbol{x}_s), (\boldsymbol{x}, y) \in \mathcal{D}_s\}$. The external features (contained in the style image) will be learned by the victim model $V_\theta$ during the training process via $\min_\theta \sum_{(\boldsymbol{x}, y) \in \mathcal{D}_b \cup \mathcal{D}_t} \mathcal{L}(V_\theta(\boldsymbol{x}), y)$, where $\mathcal{D}_b \triangleq \mathcal{D} \backslash \mathcal{D}_s$ and $\mathcal{L}(\cdot)$ is the loss function (*e.g.*, the cross-entropy).

### 4.3. Training Ownership Meta-Classifier

Since there is no explicit expression of external features and those features also have minor influences on the prediction, we need to train an additional meta-classifier to determine whether a suspicious model contains the knowledge of those features. In this paper, we train the meta-classifier $C_{\boldsymbol{w}} : \mathbb{R}^{|\boldsymbol{\theta}|} \to \{-1, +1\}$ based on the gradient of models.

Specifically, we assume that the victim model $V$ and the suspicious model $S$ have the same structure. This assumption can be easily satisfied since the defender

*Table 3.* Results on the CIFAR-10 dataset.

| Model Stealing Attack | | Model Watermarking | | Dataset Inference | | Ours | |
|---|---|---|---|---|---|---|---|
| | | $\Delta\mu$ | p-value | $\Delta\mu$ | p-value | $\Delta\mu$ | p-value |
| $\mathcal{A}_D$ | Distillation | $-10^{-3}$ | 0.32 | - | $10^{-4}$ | **0.53** | **$10^{-7}$** |
| | Diff. Architecture | 0.82 | **$10^{-12}$** | - | $10^{-4}$ | **0.95** | $10^{-7}$ |
| $\mathcal{A}_M$ | Zero-Shot Learning | $10^{-25}$ | 0.22 | - | $10^{-2}$ | **0.52** | $10^{-5}$ |
| | Fine-tuning | $10^{-23}$ | 0.28 | - | $10^{-5}$ | **0.50** | $10^{-6}$ |
| $\mathcal{A}_Q$ | Label-query | $10^{-27}$ | 0.20 | - | $10^{-3}$ | **0.52** | $10^{-4}$ |
| | Logit-query | $10^{-27}$ | 0.23 | - | $10^{-3}$ | **0.54** | $10^{-4}$ |
| Benign | Independent | $10^{-20}$ | 0.33 | - | **1.0** | **0.0** | **1.0** |

*Table 4.* Results on the 20-classes ImageNet sub-dataset.

| Model Stealing Attack | | Model Watermarking | | Dataset Inference | | Ours | |
|---|---|---|---|---|---|---|---|
| | | $\Delta\mu$ | p-value | $\Delta\mu$ | p-value | $\Delta\mu$ | p-value |
| $\mathcal{A}_D$ | Distillation | $10^{-4}$ | 0.43 | - | $10^{-3}$ | **0.61** | **$10^{-5}$** |
| | Diff. Architecture | 0.78 | **$10^{-9}$** | - | $10^{-6}$ | **0.90** | $10^{-5}$ |
| $\mathcal{A}_M$ | Zero-Shot Learning | $10^{-12}$ | 0.33 | - | $10^{-3}$ | **0.53** | $10^{-4}$ |
| | Fine-tuning | $10^{-20}$ | 0.20 | - | $10^{-4}$ | **0.60** | $10^{-5}$ |
| $\mathcal{A}_Q$ | Label-query | $10^{-23}$ | 0.29 | - | **$10^{-3}$** | **0.55** | $10^{-3}$ |
| | Logit-query | $10^{-23}$ | 0.38 | - | $10^{-3}$ | **0.55** | $10^{-4}$ |
| Benign | Independent | $10^{-24}$ | 0.38 | - | 0.98 | **$10^{-5}$** | **0.99** |

can retain a copy of the suspicious model as the victim model, based on the training set of deployed model. Once the suspicious model is obtained, the defender will train its benign version (*i.e.*, the $B$) on the original training set $\mathcal{D}$. After that, we can obtain the training set $\mathcal{D}_c$ of classifier $C$ via $\mathcal{D}_c = \mathcal{D}_{positive} \cup \mathcal{D}_{negative}$, where we have $\mathcal{D}_{positive} = \{(g_V(\boldsymbol{x}'), +1) \,|\, (\boldsymbol{x}', y) \in \mathcal{D}_t\}$, $\mathcal{D}_{negative} = \{(g_B(\boldsymbol{x}'), -1) \,|\, (\boldsymbol{x}', y) \in \mathcal{D}_t\}$, $g_V(\boldsymbol{x}') = \text{sign}(\nabla_{\boldsymbol{\theta}}\mathcal{L}(V(\boldsymbol{x}'), y))$, $g_B(\boldsymbol{x}') = \text{sign}(\nabla_{\boldsymbol{\theta}}\mathcal{L}(B(\boldsymbol{x}'), y))$, and $\text{sign}(\cdot)$ is the sign function (Hogg et al., 2005). At the end, the meta-classifier $C_{\boldsymbol{w}}$ is trained by

$$\min_{\boldsymbol{w}} \sum_{(\boldsymbol{x}, y) \in \mathcal{D}_c} \mathcal{L}(C_{\boldsymbol{w}}(\boldsymbol{x}), y). \tag{2}$$

### 4.4. Ownership Verification with Hypothesis Test

When the meta-classifier is trained, given a transformed image $\boldsymbol{x}'$ and its label $y$, the defender can examine the suspicious model simply by the result of $C(g_S(\boldsymbol{x}'))$, where $g_S(\boldsymbol{x}') = \text{sign}(\nabla_{\boldsymbol{\theta}}\mathcal{L}(S(\boldsymbol{x}'), y))$. If $C(g_S(\boldsymbol{x}')) = 1$, the suspicious model is considered as stolen from the victim. However, this method may be sharply affected by the randomness in the selection of $\boldsymbol{x}'$. In this paper, we design a hypothesis test based verification method to increase the verification confidence, as follows:

**Definition 2.** *Let $\boldsymbol{X}'$ denotes the variable of transformed images, while $\mu_S$ and $\mu_B$ indicates the posterior probability of the event $C(g_S(\boldsymbol{X}')) = 1$ and $C(g_B(\boldsymbol{X}')) = 1$, respectively. Given a null hypothesis $H_0 : \mu_S \le \mu_B$ ($H_1 : \mu_S > \mu_B$), we claim that the suspicous model $S$ is stolen from the victim if and only if the $H_0$ is rejected.*

In practice, we randomly sample $m$ different transformed images from $\mathcal{D}_t$ to conduct the pair-wise T-test (Hogg et al., 2005) and calculate its p-value. When the p-value is smaller than the significance level $\alpha$, $H_0$ is rejected.

## 5. Experiments

**Baseline Selection.** We compare our defense with the dataset inference (Maini et al., 2021) and backdoor-based model watermarking (Adi et al., 2018). Following the settings in (Maini et al., 2021), we conduct model stealing attacks illustrated in Section 2.1 to evaluate the effectiveness of defenses. Besides, we also provide the results of examining a suspicious model which is not stolen from the victim (dubbed 'Independent') for reference.

**Evaluation Metric.** We use the confidence score $\Delta\mu$ and p-value for the evaluation metric. The smaller the p-value and the larger the $\Delta\mu$, the better the defense.

**Results.** As shown in Table 3-4, our defense reaches the best performance in almost all cases. The only exception appears in defending against training with different model architectures. This is mostly because backdoor attacks are model-agnostic (Li et al., 2020a). Nevertheless, the p-value of our defense is small enough ($< 0.01$) therefore defenders can still easily identify the existence of model stealing. Moreover, our method has minor adverse effects on the performance of the victim model. Specifically, the accuracy of the model trained on the benign CIFAR-10 and its transformed version is 91.99% and 91.79%, respectively; the accuracy of the model trained on the benign ImageNet and its transformed version is 82.40% and 80.40%, respectively.

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
