# OpenReview forum: "Defending against Model Stealing via Verifying Embedded External Features"
_ICML.cc/2021/Workshop/AML — ICML 2021 Workshop AML Oral_

### Official Review · Reviewer_XMuR · 2021-06-20

**Rating:** Accept
**Confidence:** 4

**Review:**

This paper proposes to defend against model stealing attacks by embedding external features (via style transfer) into the victim model. A meta-classifier is then trained to determine whether a  suspicious model contains the external features.

This paper is written clearly that discusses the shortcomings of existing techniques. The proposed method seems to be well-motivated and effective. One thing that I did not quite understand is why the suspicious model can learn the external features of the victim model? Does is depend on model stealing attack methods?

---

### Decision · Program_Chairs · 2021-06-21

**Decision:**

Accept (Oral)

**Comment:**

A good paper that defends against model stealing attacks by leveraging external features. The paper is well-written. Please address the reviewer's comment in the final version.